# Understanding Needlestick Injuries Among Estonian Nurses: Prevalence, Contributing Conditions, and Safety Awareness

**DOI:** 10.3390/nursrep15050169

**Published:** 2025-05-12

**Authors:** Ülle Parm, Triinu Põiklik, Anna-Liisa Tamm

**Affiliations:** Physiotherapy and Environmental Health Department, Tartu Applied Health Sciences University, Nooruse 5, 50411 Tartu, Estonia

**Keywords:** needlestick injury (NSI), infection prevention, nurse awareness

## Abstract

**Background/Objective**: Needlestick injuries (NSIs) are a significant source of bloodborne infections among nurses. This study aimed to assess the prevalence, contributing factors, and awareness of post-exposure prophylaxis (PEP) among Estonian nurses. **Methods**: This prospective cross-sectional study was conducted using an electronic questionnaire in September 2024. **Results**: The majority of the 211 nurses participating in this study were females aged 21 to 75 years. Notably, 57.1% (n = 109, aged 43.9 ± 12.2) had experienced an NSI in the past decade. Most injuries occurred during sharps’ disposal (33%) and with syringe needles (72%). Among those injured, 84% washed the area with water and soap, 80% used alcohol-based disinfectants, and 69% reported the incident. However, 20.6% did not report due to perceived insignificance or lack of follow-up actions. Additionally, 14.7% were unaware of the reporting requirement, and 8.8% did not know who to report to. **Conclusions**: Improved training and reporting practices are essential to reduce NSIs among nurses.

## 1. Introduction

Healthcare workers, especially nurses, are at increased risk of exposure to the blood, tissue, or other bodily fluids of infected patients, which, among other factors, may cause needlestick infection [1,2,3]. For example, in North America, millions of healthcare workers use needles in their daily work, and therefore, their risk of incurring needlestick injuries (NSIs) is of high concern [2]. The risk certainly varies from country to country, but according to a review based on 87 studies, comprising a total of 50,916 respondents, and data from 31 countries, the annual prevalence of NSIs among healthcare workers is 44.5%. In Europe, studies have estimated over one million NSI cases per year [4] and that most healthcare staff have been exposed at least once during their professional lives [3]. Although more than twenty bloodborne diseases have been identified, the three most crucial for healthcare workers are hepatitis B, hepatitis C, and HIV infection [1,2,5]. Various countries have established guidelines to help healthcare facilities manage NSIs, provide guidance on how to prevent hepatitis infection, and when to start post-exposure prophylaxis (PEP) for HIV [1,2,6]. In accordance with the regulation of the Government of the Republic of Estonia [7] and the EU directive, guidelines [8] have also been developed in Estonia, which determine the obligations of both the employer and employee when implementing precautionary measures. After a contact incident occurs, and the injury site (mucous membrane) is properly cleaned, the incident must be reported to an authorized person who determines the risk of infection and the need for PEP. Although the procedure is similar in the USA [1], Estonian healthcare professionals follow the guidelines of a health authority.

Regarding NSIs, estimations indicate a 0.2–0.5% risk of HIV, 0.5–10% risk of hepatitis C, and 5–40% risk of hepatitis B infection [1,2,9,10]. For instance, between 1981 and 2010, 143 cases of suspected HIV infection were reported by healthcare workers in the United States, 57 of whom were diagnosed with HIV [2]. The U.S. Centers for Disease Control and Prevention (CDC) and several states have developed guidelines for the rapid initiation of antiretroviral therapy to prevent infection [2,10,11,12]. Of the infections associated with NSIs, hepatitis B is the most common. Vaccination has been shown to significantly reduce the risk. After an exposure event, immunoglobulin is administered depending on the level of immunity, followed by a vaccine, if necessary, within the first few hours [2]. Unfortunately, there is no vaccine or specific PEP treatment against this disease [1]; however, more general prophylactic treatments are available [10].

Certain practices can increase the risk of NSIs, including not using or misusing safety equipment, not following universal precautions, resheathing needles, performing high-risk procedures, inappropriate collection of contaminated waste (especially sharps) when removing intravenous cannulas, and non-safety syringes and other use of sharp devices [2,5,13]. Lack of work experience also increases the risk of injury. For example, in studies conducted in Malaysia, involving 316 medical students, as many as 19.9% [14] reported an NSI, compared to 21.3% in Nepal [15]. The risk of exposure among healthcare workers also depends on the type of syringe (needle), the frequency of injury, the type of microbes in the patient’s blood, and their vaccination history [2].

It is well established that NSIs are one of the main occupational accidents among healthcare workers [3]. Most occupational bloodborne infections in healthcare workers occur through accidental punctures [1]. However, the exact number of NSIs described above is unknown, as many go unreported [2]. For example, according to a survey of 1010 nurses in Iran [16], 57.4% had experienced NSIs (including some recurrent cases), but only 10.2% reported them to their hospital infection control units. In the U.S., approximately 600,000–1,000,000 NSIs have been estimated per year [17], half of which go unreported [2]. In Estonia, the real needlestick frequency and its causes remain unknown. Only those in which preventive treatment is applied are registered as occupational accidents [7].

Although PEP treatment is successful (in HIV and hepatitis B), the reasons for non-reporting may be related to healthcare workers’ lack of awareness of the risks or appropriate instructions at the workplace or fear of disclosure among colleagues [18]. In the study in Iran, being too busy with work was identified as the most common (27.6%) reason for non-reporting [16]. Studies conducted in different countries have highlighted frequent non-reporting, for various reasons [14,16,19,20,21]; however, non-reporting data in Estonia are lacking. Corresponding information about Estonia would help clarify our possible bottlenecks and possibly save lives.

In national and international guidelines, the term NSI is used when the cut or prick is made with a needle previously used in a patient, resulting in possible exposure to blood and bodily fluids [9,10,16,21,22]. NSI occurrence has also been identified during drug preparation or before procedure [16], which is not infectious. The aim of the study was to identify the prevalence of NSIs (including not infectious or safe ones) among nurses, the conditions that contribute to their occurrence, the activities that follow an infectious injury, and awareness of the issue.

## 2. Materials and Methods

This prospective cross-sectional study was conducted using an electronic questionnaire in the LimeSurvey environment (www.limesurvey.com, LimeSurvey GmbH, Hamburg, Germany) (accessed on 11 March 2025) over four weeks in September 2024. The study received approval from the Research Ethics Committee of the University of Tartu. The link to the questionnaire was distributed with the assistance of the Estonian Nurses Union, which includes approximately 4000 nurses. As approximately 9000 nurses [23] practice in Estonia, 96% of whom are females [24], the survey was also promoted through the social media channels (Facebook and Instagram) of Tartu Applied Health Sciences University.

In the questionnaire, we sought to determine the actual number of needlestick cases among nurses (including the so-called safe sticks, which do not involve contact with blood and bodily fluids), the factors contributing to the occurrence of NSIs, nurses’ behavior after infectious NSIs involving exposure to blood and bodily fluids, and their awareness of the possibilities of preventing bloodborne diseases. Due to sensitive data and confidentiality, the nurses were not asked about their place of work (establishment or department) or even the county in which they work. In the LimeSurvey survey environment, the IP addresses of the respondents’ computers were not recorded.

We adopted a validated and standardized questionnaire used in Iran [16], which was significantly modified to meet local guidelines. The questionnaire had been piloted using six nurses as respondents. Subsequently, and after consultation with the hospital’s infection control doctor, the knowledge section was simplified. The first part of the questionnaire consisted of questions on demographic data: gender, age, length of service, immunity related to hepatitis B and C, and completion of NSI training. In contrast to the original questionnaire, we added a question on the history of hepatitis B and C and the testing of hepatitis B antibody counts. In the second part, we sought to identify the number of syringe needle incidents (cases involving and not involving contact with blood and bodily fluids separately), departments, conditions surrounding the incident, and the type of syringe involved over the past ten years. Unlike in the original questionnaire, we did not ask about the presence of needlesticks during recapping, as this is not performed in Estonia. We also omitted the questions about post-needlestick activities, as immunoglobulin injections and antibody titration are not available in Estonia without a doctor’s prescription. As regards “infectious needlesticks”, we sought to determine the follow-up and, in non-reporting cases, the reason for non-reporting. In the third part, we targeted the nurses’ informed actions after a needlestick incident and ways to prevent bloodborne diseases (HIV and hepatitis B and C).

Questionnaires with only the first part completed (n = 96) or with incomplete responses in the second part (n = 89), as well as the single questionnaire in which the respondent indicated an age of 222 years, were excluded from the data analysis. Failure to answer the last part was not a criterion for exclusion from the survey.

For statistical analysis, SigmaPlot for Windows version 11.0 (GmbH Formation, Erkrath, Germany) was used. Regarding descriptive statistics, proportions are reported as means ± SD. Continuous data and proportions were compared using a *t*-test or Mann–Whitney test and chi-square or Fisher exact tests.

## 3. Results

### 3.1. Demographic and NSI Data

The majority of the 211 participants in this study were females aged 21 to 75 years. In the past ten years, 152 (72.04%) nurses reported needlestick injuries. This study also included “safe” NSIs, cases in which an infection occurred without being in contact with the patient’s blood or bodily fluids (before needle use). Those that occurred upon contact with the patient’s liquefied fluids were designated as “real” NSIs, as they posed a risk of infection transmission. Background data for all study participants and data on stab injury comparing the two groups are presented in Table 1. In total, 102 participants reported safe and real NSIs (48.34%; mean age 43.97 ± 12.45 y), whereas 49 respondents reported no needlestick injuries (23.22%; mean age 45.71 ± 12.1 y; min 22, max 65 y).

Safe NSI—no contact with the patient’s bodily fluids or blood; real NCI—possible contact with the patient’s bodily fluids or blood, resulting in the risk of infection transmission. There were no statistical differences between these two groups. Regarding needle type, the lancet was most frequently cited, with three times.

In addition to the departments presented in the table above, safe needlestick injuries also occurred in the ambulatory blood collection office, private care center, geriatrics, hematology (n = 2), emergency medicine (n = 2), radiation therapy, school nursing (n = 6), neurology (n = 2), oncology (n=3), dermatology, psychiatry (n = 4), rehabilitation (n = 3), gynecology, and nursing departments. Real NSIs occurred in kindergarten (n = 9), nursing homes (n = 6), ambulatory blood collection office, private care, geriatrics department, dentistry (n = 3), hematology (n = 2), nursing home (n = 2), hospice, emergency room (n = 2), school nursing (n = 3), laboratory, neurology, oncology (n = 2), orthopedics, procedure room, psychiatry (n = 2), sexual health office, rehabilitation (n = 2), vaccination office, and nursing care (n = 3).

The estimation of the actual number of NSIs proved difficult, as cases with more than two incidents were specified with the term “many” or ranges such as 5–7, 8–10, etc. Respondents often stated that they did not remember their last needlestick injury. Regarding the conditions surrounding NSI occurrence, the questionnaire responses allowed us to determine whether the participants considered the incidents dangerous or safe. The results presented below refer to the responses of the interviewees.

### 3.2. The Behavior of Nurses After Real NSIs and Opinions on the Actions Needed

In the second part of the questionnaire, the nurses described their behavior after a real NSI (n = 109), and the third part was completed by all those who agreed to share their views regarding the actions needed after NSI occurrence (n = 134). The results are presented in Figure 1. Notably, 55.1% of NSIs were reported properly within half an hour, with a further 22% of cases reported within 12 h. Although the proposed actions were similar, statistical differences were observed between nurses’ actions in practice and their views. Activities such as washing the wound with running water and soap (*p* < 0.001), using an alcohol-based or alcohol-free cleanser (*p* < 0.001, *p* = 0.006), and applying pressure to the area to encourage bleeding (*p* < 0.001) were performed more often than perceived necessary. At the same time, the respondents considered the following actions necessary more often than in practice: reporting in the prescribed manner (*p* < 0.001), ensuring free flow of blood (*p* = 0.031), drying the wound and plastering (*p* = 0.001), and using alcohol to clean the wound due to lack of water (*p* = 0.031).

### 3.3. Reasons for Not Reporting NSIs

Notably, 31.19% (n = 34) of the study participants did not report real NSIs (n = 109) according to the prescribed procedure. Non-reporting reasons are presented in Table 2. In addition to the reasons indicated in the table, seven participants provided the following explanations: the patient was a child with no medical history; the patient did not suffer from a bloodborne disease; they assessed the situation as safe; lack of protocol regarding to whom and where they should report; they recorded in writing the date of the injection and the name of the person to whom they administered the injection; notification was not customary at the family doctor’s center; and they felt ashamed. Some respondents also stated that they reported once, but upon seeing no further action taken, they did not repeat it the second time.

### 3.4. Nurses’ Views on the Prevention of Major Bloodborne Diseases

The completion of the last part of the survey was voluntary; in this section, the nurses’ knowledge of and opinions about NSI follow-up (see Section 3.2) and the possibilities of prevention/treatment of the three main bloodborne diseases are presented (Table 3). This part was completed by 63.51% of all participants in the study (n = 134: without NSIs n = 26; with NSIs 108; with real NSIs n = 74; with safe NSIs n = 104; with both NSIs n = 71).

## 4. Discussion

The results of this study revealed that two-thirds of the respondents (73%) had an NSI in the last ten years, while more than half of the respondents (52%) considered them dangerous for the transmission of infection, as they involved exposure to patient’s blood and bodily fluids (real NSIs). In one-third of those who had experienced a real NSI, the incident occurred more than twice, providing evidence for the common occurrence of these injuries [4,13]. According to the data of the Labor Inspectorate [25], occupational accidents involving syringes and needles occurred between 2014 and 2022 among 58 healthcare workers (5 of whom were physicians); however, the number of nurses in the above finding is unknown. Although more than a fifth of our participants have had no history of a needlestick injury in the last 10 years, according to the data reported in 31 countries, all healthcare workers had at least one in the last year. Global prevalence was slightly lower than that reported by us (44.5%) [4].

According to the results of a previous study [4], a syringe (hypodermic) needle was the cause of NSIs in 46% of cases, followed by a suture needle (20%) and an IV cannula (26%); in comparison, in the current study, needlestick injuries occurred more often with a syringe needle and less often with a suture needle. The role of different needles in causing NSIs varies among studies. For example, in a study carried out in Iran, although a syringe needle was the most common cause, more than a fifth of the cases were caused by a butterfly needle [16]. Needles with so-called safety devices (i.e., safety-based protection mechanisms) have been developed to reduce NSIs; however, in the current study, NSIs were also reported with their use. Additionally, previous studies have shown that their use even increased the number of needlestick injuries (from 1.9 to 2.2 per 100 healthcare workers) [26]. This may be attributed to the department with which the study participants were affiliated. Although we based the questionnaire on the survey by Joukar et al. [16], many of those who responded to the questionnaire did not find a job listed in the questionnaire that matched their practice. Thus, the jobs of the participants in this study were more diverse, and NSI occurred more often in the internal department.

According to the responses of the participants regarding the activity causing NSIs, the results indicating a real or safe NSI were inconclusive. For example, in safe needlestick injuries, it was also reported to occur during blood collection, injection, or throwing sharp objects into a container. Real NSIs were found to have occurred during preparation for the procedure. As this often occurs among nurses during recapping [13], the process is not recommended [10], nor is it in Estonia. In our study, NSIs occurred during the disposal of sharps in containers in a third of the cases, while in nearly a quarter, they were associated with patients’ sudden movement. The latter can be explained by the fact that a considerable number of NSIs occurred in kindergarten or school, where there is likely a more restive contingent, although no needling occurred in the pediatric ward. Previous studies have also reported restlessness or patients’ sudden movement as the cause of needlestick injuries [16,27]. Based on the results of a previous study, which includes data from NSI-related documents submitted by healthcare workers [27], the chance of NSI occurrence is five times (OR = 4.84; 95% CI 1.71–13.7) higher when suturing a wound than other activities.

More than 10% did not use gloves; however, they were still more commonly used than reported in other studies [13]. Wearing proper protective gear, including gloves, is crucial to prevent infection [10], and in some cases, the use of double gloves is also recommended [2]. However, the use of protective gear does not guarantee 100% protection [28]. Analogous to the previous studies [13,16,27], NSIs occurred most frequently during the morning shift. This is likely the time during which a larger number of medical procedures are performed on a larger number of patients.

The most frequent actions during the first half hour after a needlestick injury were holding the relevant area under running water and washing with soap; treating the wound with an alcohol-containing cleaner; and wound cleaning and plastering. Other studies also identified holding the finger (the damaged area) under water as the first step after an NSI [16,22]. In agreement with our findings, activities such as applying pressure on the site to promote bleeding [16] or squeezing out blood have also been reported [22]. Guidelines typically recommend immediately washing the area with soap and water, and the claim regarding the need to use an antiseptic or skin washes has not been proven. However, some guidelines recommend that after washing the wound with soap and water allowing free bleeding, a cleaner with 70% alcohol should be used [10].

According to the World Health Organization, 16,000 healthcare workers contracted hepatitis C every year due to NSIs, compared to 66,000 with hepatitis B and 100 with HIV infection [4]. The risk of any infection depends on the type of needle, the severity of the NSI, the type of pathogen, the availability of PEP, and vaccination status [2]. Hepatitis B has a prophylactic vaccine, and all healthcare workers should be immunized [10]. PEP with immunoglobulin and active immunization depends on the recipient’s vaccination status [2,10]. The immunization status of hepatitis B in our study was relatively good but not excellent, but nurses’ awareness of the general prophylaxis and the treatment of these diseases, as well as the possibilities of PEP, were deemed unsatisfactory. However, nurses’ awareness was relatively better than in the study conducted in India [22], where only a tenth of nurses knew about the transmission of these three diseases, especially hepatitis C, through needlestick injuries.

Although the CDC and various countries have developed guidelines for the prompt initiation of antiretroviral therapy to prevent infection [2,9,10,12], and Estonia is no exception, less than half of the participants in this study had been trained in NSIs. Although the relevant guidelines are presented to each staff member when they start working in a hospital, this is not always the case throughout the health system, and adequate recurrent training is lacking. More than a fifth of those who suffered an NSI did not know that it had to be reported or did not know to whom it should be reported. The non-reporting problem has also been identified in studies conducted in other countries [22,29]. The reasons for non-reporting, as well as knowledge about the possibilities of bloodborne disease prevention, indicate insufficient data on the relevant topic. Better knowledge and the courage to report a needlestick injury facilitate PEP planning for those at risk, thus helping to prevent possible infection. The high prevalence of NSIs reported in the survey covering 31 countries highlights the need to improve occupational health services and needlestick education programs worldwide [4].

This study has several limitations. For example, the minimum number of participants for adequate study power was 369, and 396 participants responded at least partially to the survey, but only the responses of those who correctly (n = 211) completed the required part of the questionnaire were included in the data analysis. We are therefore aware that this study lacks generalizability due to an insufficient number of respondents. We considered “safe injections”, which involve no contact with patients’ blood or other bodily fluids. Not all nurses were found to be able to distinguish between a real and safe syringe, although we provided precise explanations of the terms. Additionally, we could not report the number of NSIs because some had experienced recurrent incidents and could not recall them accurately. It is also worth noting that NSIs (including non-infectious ones) can cause psychological distress, leading to anxiety [30] and absenteeism. Unfortunately, this was not addressed in our study. The findings provide guidance for future questionnaires and studies. Despite the shortcomings, we found that the questionnaire in this study helps to provide an overview of the state of NSI occurrence, which provides important insights for curriculum development and training of professionals.

## 5. Conclusions

NSIs are common, and in most cases, they occur either during injections, blood draws, or disposal of sharp objects in containers. The solution to reducing NSIs involves establishing clear and coherent policies across medical and healthcare institutions, reflecting key prevention guidelines. Although institutions have guidelines for addressing exposure incidents, developed by their infectious disease committees, they likely lack sufficient attention and are not frequently consulted, for example, during in-house training. The actions taken after an NSI varies, and the knowledge about the prevention of bloodborne diseases is poor. A third of the participants in this study did not report their NSI, which did not allow them to perform PEP. Nurses should be encouraged to report needlestick incidents and should be aware of bloodborne disease prevention options.

## Figures and Tables

**Figure 1 nursrep-15-00169-f001:**
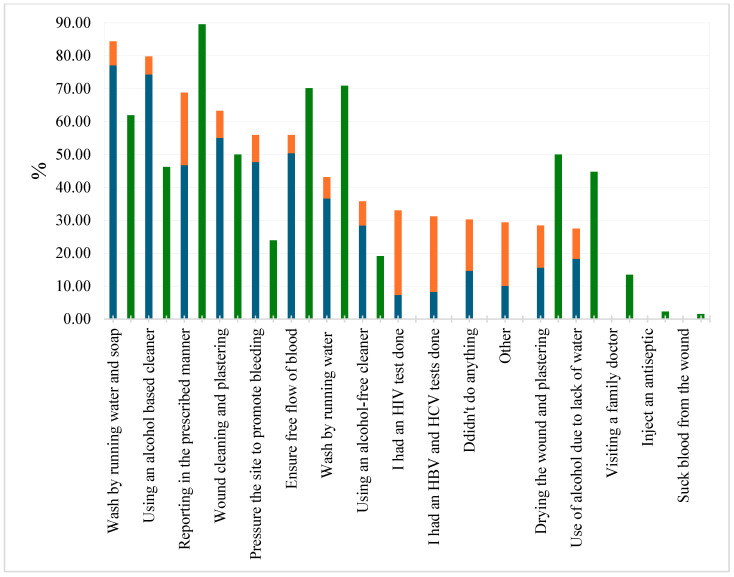
Nurses’ actions after NSI occurrence. Actions after real NSIs (n = 109) are presented in blue (within 0.5 h after an NSI) and orange (within 12 h). The opinions (n = 134) on the actions required are presented in green.

**Table 1 nursrep-15-00169-t001:** Gender, age, and NSI data.

Characteristics		All Nurses	Safe NSIs	Real NSIs
N = (%)		211	155 (73.5)	109 (51.7)
Gender: female (%)		97.63	97.42	97.25
Age (mean ± SD)		44.81 ± 12.32	44.63 ± 12.6	43.88 ± 12.17
Immunological status: hepatitis B (%)	I have had this disease	1.90	1.94	1.83
	I am not vaccinated	6.16	7.1	6.4
	Vaccinated <5 y ago	15.64	14.84	19.27
	Vaccinated 5–10 y ago	12.8	11.61	11.93
	Vaccinated >10 y ago	27.49	26.45	23.85
	Antibodies have not been checked	6.16	7.74	8.26
	Antibodies have been checked within 5 y	17.54	16.77	18.35
	Antibody levels are up to the mark	9.0	9.68	5.5
	Few antibodies—I plan to vaccinate	3.79	3.87	3.66
	Few antibodies—do not plan to vaccinate	0.95	1.29	1.83
Immunological status: hepatitis C (%)	I have had this disease	0.95	1.29	0.92
	I do not know if I have had this disease	29.38	32.26	33.03
I have been trained in NSI		45.97	44.52	44.95
The number of needlesticks in the last ten years (%)	Once	41.71	33.55	33.03
	Twice	36.49	33.55	22.94
	More than twice	42.65	32.26	36.7
Occupational ward, where the needlestick injury occurred (%)	Pediatrics	3.32	4.52	0
	Emergency department	17.54	14.84	12.84
	Internal medicine	29.38	24.52	22.02
	Obstetrics	2.37	2.58	0.91
	Surgery	14.69	11.61	11.93
	Operation room	5.21	4.52	3.67
	Intensive care unit	9.95	8.39	7.34
	Family doctor/nurse center	21.33	18.71	14.68
NSI time (%)	Morning/day		66.45	67.98
	Evening		27.74	30.28
	Night		12.9	15.59
Using gloves during a needlestick procedure			80.65	88.07
Type of needle (%)	Syringe needle (nature needle)	56.87	74.84	71.56
	Suture needle	3.79	5.16	6.42
	Winged butterfly (winged steel needle)	6.63	8.39	11.01
	IV catheter stylet	13.27	17.42	22.02
	Needles with safety-engineered devices	2.84	3.23	5.5
	Other	8.53	11.61	13.76
Action associated with injury (%)	Upon preparation	26.54	36.13	18.35
	During taking blood	10.43	14.19	17.43
	During injection	9.0	10.97	14.68
	Upon pulling out	9.48	11.61	16.51
	When suturing a wound	2.37	3.23	4.59
	During sharps’ disposal	17.54	23.23	33.03
	Patient’s sudden movement	14.22	18.71	23.77
	Accidental prick from others	0.95	0.65	1.83

**Table 2 nursrep-15-00169-t002:** Reasons for not reporting NSIs.

Did not know that I need to inform	14.71
Did not know who to inform	8.82
I was too busy, I forgot	17.65
I was afraid of the negative impact on my work	8.82
I was afraid of losing my job	2.94
It’s not that important	20.59
After reporting, nothing is done	20.59
I had the laboratory done myself—was negative	2.94
Other	20.59
Didn’t want to comment	17.65

**Table 3 nursrep-15-00169-t003:** Nurses’ opinions on the prevention or treatment of major bloodborne diseases.

Opinions	Hepatitis B	Hepatitis C	HIV
The disease can be prevented by vaccination.	96.27	25.37	4.48
Lack of treatment	8.21	12.69	9.70
Treatment does not allow the pathogen to be completely defeated.	38.06	48.51	76.12
The treatment is effective and the person recovers completely.	17.91	30.60	2.99
Preventive treatment prevents the development of the disease if it is started as soon as possible.	39.55	ND	58.96
After an NSI, there is no preventive treatment for this disease.	17.91	24.63	11.94

ND—no data.

## Data Availability

We are aware that the data is anonymous, but the University does not currently have a repository or a way to submit open data. Therefore, the data is stored on a secure server at the University and we can provide access to it in accordance with the University’s internal policy in case of a formal request.

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
