# Peer review of "Understanding Needlestick Injuries Among Estonian Nurses: Prevalence, Contributing Conditions, and Safety Awareness"

_nursrep, 2025, doi:10.3390/nursrep15050169_

Round 1

Reviewer 1 Report

Comments and Suggestions for Authors

The article " Needlestick Injuries among Estonian Nurses “is a study on nurses' knowledge of appropriate needlestick reactions.  

Abstract:

Please revise the abstract. The conclusion part of the abstract says,” Results indicated a significant gap in nurses' knowledge of hepatitis B, C, and HIV prevention and treatment. Enhanced training is crucial.” There is no data about these diseases in the results section of the abstract.

Introduction:

Line 68: “are approximately 600,000 – 100,000 NSIs per year.” Please rewrite it.

Lines 72- 82: These are most appropriate for the method and material section, I think. The same information was stated in the first part of the method and material section.

Results:

The study population is a big concern for me.

Filling the questionnaire by 211 nurses among the 9000 nurses in Estonia is a big weakness of the study. The questionnaire was filled out by only 2.3% of nurses in the country.  Is it enough to measure the knowledge of Estonian nurses about the study subjects?

The other big concern is the gender of the subjects of study. 97.4% of those who filled out the form were women. What is the proportion of women in the nurse population in Estonia? I think samples are not normally distributed.

Discussion:

The discussion is not easy to follow in some parts. Please revise it, and start it with the main finding of your study.

Lines 208 and 213: Please use “current study” instead of “our work” in the text. 

Comments on the Quality of English Language

I think English needs minor revise. The discussion part is not easy to follow in some parts. 

Author Response

Dear reviewer, you will find our comments and answers in WORD document. Best wishes!

Reviewer 2 Report

Comments and Suggestions for Authors

Review for the manuscript titled Needlestick Injuries among Estonian Nurses, submitted in Nursing Reports.

The topic of needlestick injuries (NSIs) among nurses is highly important as it directly impacts occupational safety, public health, and healthcare system efficiency. Also, it is well-known that nurses are at a high risk of NSIs due to frequent contact with needles and sharp instruments. NSIs can expose nurses to serious infections such as hepatitis B (HBV), hepatitis C (HCV), and HIV, which the author well described in the manuscript. However, even non-infectious NSIs can cause psychological distress, leading to anxiety and lost work time, so this also should be mentioned in the manuscript. Therefore, I recommend a major revision.

Title

The manuscript title is general and corresponds to the title of a review manuscript, not a manuscript presenting the results of an empirical study. Therefore, I suggest its correction to make it more informative. For example: Understanding Needlestick Injuries among Estonian Nurses: Prevalence, Contributing Conditions, and Safety Awareness, or Prevalence, Contributing Factors, and Post-Injury Actions Related to Needlestick Injuries Among Estoninan Nurses: A Study on Occupational Safety and Awareness

Abstract

The abstract is informative and has a sufficient number of words. However, the conclusion does not meet the objectives of the study. I propose to correct the sentences written in lines 16, 17, and 18 with the indication of the prevalence rate, the most significant factor contributing to NSI, and a proposal for measures at the end.

Introduction

The authors use various terms like “healthcare providers, healthcare workers, health professionals” for better readability; I suggest using one term and sticking to it throughout the text.

It is necessary to correct the sentences in lines 25 to 29. They refer to the same reference source [4]. It would be logical if the second sentence, which refers to data from 31 countries, is listed first, followed by the European data.

In line 62, the word “So” is unnecessary. The last paragraph is not clear enough. For example, in line 69, the authors state, “as described in the instructions,” I do not understand which instructions. Also, the term in line 81, “At the same time,” should be replaced with “While”.

I believe the introduction does not cover all the reasons why it is important to investigate the prevalence, contributing factors, and nurses’ awareness of NSI. Therefore, I propose to write a paragraph about the consequences of NSI, which do not refer to serious infections but to psychological distress with the resulting consequences. Thus, the introduction would have the necessary funnel approach.

Material and Methods

The authors state that they used a validated and standardized questionnaire used in Iraq as a research instrument. However, the reference [16] they refer to indicates that it is the Iranian questionnaire. Such a statement must be checked and corrected. Then, the authors further state that they made the modification for their research purposes. It is necessary to describe in more detail and indicate which modifications were made and why.

Also, I think the paragraph from lines 117 to 121 should be corrected. More clearly describe the inclusion criteria and check what “one who indicated an age of 222” refers to.

Results

The results are presented through three tables and one figure. I propose to replace the terms participants, subjects, and the like with the term nurses.

Table 1 is titled Demographic and NSI data. Who and what demographics? It is necessary to be more precise.

The % sign must be moved to the main line, and absolute numbers (n), not only relative (%), must also be specified.

A general objection to the results and statistical analysis is the absence of regression analysis. Namely, one of the study’s aims was to determine the situations that contribute to the occurrence of NSI - there is no such thing in the results.

Discussion and conclusions are drawn coherently and supported by the listed citations, and limitations are described.

The cited references are mostly recent publications (within the last five years) and relevant.

Overall assessment

Overall, this manuscript provides a valuable contribution to the nursing safety field. Strengthening the clarity and expanding the methodology would further enhance its impact. With major revisions, the study has strong potential for publication in Nursing Reports.

Author Response

(The authors gave the same response as above.)

Reviewer 3 Report

Comments and Suggestions for Authors

Dear everyone, good afternoon, I attach the file of suggestions from me as reviewer of the manuscript.

Author Response

(The authors gave the same response as above.)

Reviewer 4 Report

Comments and Suggestions for Authors

dear Authors, thank you for your paper. did you perform a comparison with other countries? what about results/wy not? 2) you shoul use only references with 5 years; some references are too old

Author Response

(The authors gave the same response as above.)

Reviewer 5 Report

Comments and Suggestions for Authors

Why was the education variable category not included in the demographic data?

Could it be assumed that educational attainment would have an impact on the results in the study?

Author Response

(The authors gave the same response as above.)

Round 2

Reviewer 1 Report

Comments and Suggestions for Authors

In my point of view, article is ready for publishing. 

Reviewer 2 Report

Comments and Suggestions for Authors

Dear Authors,

Thank you for your careful and thoughtful revisions to the manuscript. I appreciate your effort in proofreading English and addressing my comments, which improve the overall quality of your manuscript.

You have successfully revised key sections of the manuscript, including the title, abstract, introduction, material and methods, and results, in line with the suggested recommendations. The manuscript is now clearer, more coherent, and aligned with the journal’s standards.

While a more advanced statistical analysis was suggested and ultimately not implemented, I acknowledge your explanation and find it reasonable within the context of your study. The manuscript still contributes to nursing research, addressing a relevant and timely issue.

I support the publication of this revised version and commend your research and analysis of this important topic.

Sincerely,

Reviewer 4 Report

Comments and Suggestions for Authors

thank you for your work!